# GraB: Graph Benchmark for Heterogeneous Graph Clustering

**Malik Stær Knudsen**
Aarhus University
201905297@post.au.dk

**Laurits Almskou Brodal**
Aarhus University
201908901@post.au.dk

**Peter Kristoffer Peczalski**
Aarhus University
201909494@post.au.dk

**Atefeh Moradan**
Aarhus University
atefeh.moradan@cs.au.dk

**Davide Mottin**
Aarhus Univeristy
davide@cs.au.dk

**Ira Assent**
Aarhus University
ira@cs.au.dk

## Abstract

We introduce GraB, a benchmark for graph clustering with unique characteristics. Our graphs are at the same time heterogeneous, i.e., include different types of nodes and node attributes, and comprise overlapping clusters, i.e., a node may belong to multiple clusters. We empirically show the arduous characteristics of the datasets; GraB is available at https://github.com/AU-DIS/GraB.

## 1 Introduction

*Graph clustering* detects groups of nodes in a graph by analyzing relationships among nodes. In social networks, people subscribe to different groups [1]; in co-citation networks, papers belong to various research areas [2], in protein-protein interaction networks clusters represent proteins complexes [3].

*Evaluating graph clustering algorithms* is a challenging task, which requires ground truth information, using synthetic data [4] or real-world data collected from sources like Facebook [1]. Synthetic data provides controlled experiments, but may not necessarily reflect all properties present in real-world data. *Real-world graphs* are usually sparse and may also include descriptive *attributes* for nodes.

*Scale* is also important for benchmarking, but larger graphs ($> 10k$ nodes) for overlapping graph clustering are typically only available for homogeneous graphs [1, 5] of the same type nodes only. We lack benchmarks of larger *heterogeneous* graphs, where nodes may belong to different types, e.g., metabolic networks of chemical components and chemical reactions contain two types of nodes [6].

Most existing benchmarks focus on assigning nodes to a single cluster, to evaluate non-overlapping graph clustering. Some more recent approaches, however, study *overlapping* graph clustering, such as people or entities belonging to multiple groups. In this work, our focus is on providing benchmarking for such overlapping graph clustering as well.

Finally, benchmarking against the same few datasets from a few domains may bias the evaluation and entail misleading results and conclusions.

In the worst-case, this scarcity limits research progress in the area, as we lack knowledge about algorithms' performance for other types of graphs [5].

**Related work.** Table 1 reviews the main related work on benchmarks for graph clustering.

*Synthetic graph benchmarks* sample graphs from a predefined distribution: the Lancichenecchi-Fortunato-Radicchi (LFR) benchmark [4], one of the most popular synthetic graph benchmarks, generates overlapping clusters, but without any attributes. acMark [7] extends LFR with attributes.

*Real graph benchmarks* typically contain several real-world graphs. Notably, SNAP [8] includes several graphs with different characteristics, but no attributed graph in SNAP has overlapping clusters. NOCD [1] has a number of small graphs (<1000 nodes), including attributes and overlapping clusters.

Knudsen M. et al., GraB: Graph Benchmark for Heterogeneous Graph Clustering (Extended Abstract). Presented at the First Learning on Graphs Conference (LoG 2022), Virtual Event, December 9–12, 2022.

No prior benchmark for graph clustering considers overlapping clusters on heterogeneous attributed graphs, and provides the size to assess scalability. We fill this gap with *GraB* (Graph Benchmark).

| Benchmark | Overlapping | Attributed | Heterogeneous | Real Data |
|---|:---:|:---:|:---:|:---:|
| LFR [4] | ✔ | ✘ | ✘ | ✘ |
| acMark [7] | ✔ | ✔ | ✘ | ✘ |
| SNAP [8] | ✔ | ✘ | ✘ | ✔ |
| NOCD [1] | ✔ | ✔ | ✘ | ✔ |
| GraB | ✔ | ✔ | ✔ | ✔ |

**Table 1:** Main benchmarks for graph clustering and their characteristics.

## 2 Dataset

Our desiderata is to obtain novel graph benchmarks with real overlapping clusters, node attributes, and heterogeneity. We select the movie domain which provides widely available data with nodes of different types, attributes and multiple group memberships. Multiple roles (e.g. actor, director etc.), make movie graphs heterogeneous. Also, movie descriptions are generally easy to understand, rendering the results of an algorithm more interpretable. We obtain GraB by integrating IMDb information into DBpedia.

**GraB construction.** DBpedia [9] is a rich knowledge graph extracted from Wikipedia. Each node in DBpedia is a Wikipedia page with attributes of varying detail. DBpedia contains a wealth of movies with attributes, such as movie length, however, lacks the movie's genres that could represent natural clusters. To this end, we extract genres from IMDb [10], a complete repository of movies, actors, and ratings. We extract movies from DBpedia, match them with the corresponding description in IMDb, and add the genre and plot keywords. We also extract people, such as actors, producers, directors, editors, and writers connected to each movie. To assign a genre to an actor we devise three strategies described in Section 2.2.

To obtain a connected graph, we perform a breadth-first search from a few nodes of well-connected movies and actors (e.g., Brad Pitt). All people in the dataset only have edges to movies, and all movies only have edges to people. Movies are not directly connected to one another, same for people.

The dataset naturally extends to a large number of connected movies and people, which results in 9 367 movies, 4 832 actors, 1 915 writers, 1 617 producers, 1 582 directors and 543 editors.

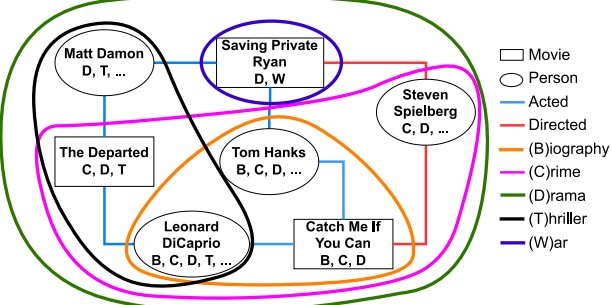

**Figure 1:** GraB excerpt: node (shapes) and edge (color) types, overlapping clusters (enclosing colored lines).

All nodes with type person are connected to at least two movies, and fewer than 500 movies have only one edge. We exclude person nodes with a single edge because they would only inherit the same genres from that movie (as in Section 2.2), and trivially belong to the same clusters. Figure 1 shows an excerpt of GraB with node and edge types, and an illustration of the cluster affiliations of nodes.

### 2.1 Attribute Selection

**Movie genres.** The genre of a movie, which naturally determines clusters, is absent in DBpedia. We include additional data from IMDb, to obtain the genre for the movies [10][1].

**Attribute inconsistency.** Some attributes in the DBpedia graph have inconsistent data formats (e.g., strings and integers) and are not directly comparable. We manually convert attributes in the same format, e.g., all currencies to integers. The nodes in the graph have a heterogeneous set of attributes as movies and people differ in type and description.

**Selected attributes.** Some attributes do not contain useful information for graph clustering. Attributes, such as the size of the picture on the Wikipedia page are discarded. We also discard attributes having

---

[1]We use the attributes *primaryTitle* and *runtimeMinutes* to match a movie in DBpedia with one in IMDb.

| | Cluster statistics | | | | | Overlap size | | | |
|---------|------|------|----------|---------|-------|-------|------|--------|--------|--------|
| Dataset | Avg. | Std. | Smallest | Largest | CN | NN | 1 | 2 | 3 | 4 |
| Full | 3 817 | 3 367 | 585 | 12 954 | 5.7% | 18.5% | 100% | 89,82% | 72,79% | 42,61% |
| Min | 2 558 | 2 949 | 204 | 11 281 | 3.2% | 7.8% | 100% | 76,59% | 51,55% | 19,92% |
| Top 3 | 2 883 | 3 220 | 299 | 12 277 | 14.6 % | 34.4% | 100% | 89,83% | 72,80% | 28,84% |

**Table 2:** Cluster statistics for each dataset (Full, Min, Top-3): average cluster size (Avg.) and its standard deviation (Std.), Smallest, Largest cluster sizes; percentage of disjoint nodes over all clusters (CN) and in total nodes (NN); percentage of nodes in at least 1, 2, 3 or 4 clusters (Overlap size).

only a single value or IDs of nodes. On the other hand, we retain unique numerical attributes with a specific meaning, such as the movie budget. To this end, we scrupulously inspect each attribute individually.

**Textual attributes.** In addition to the attributes extracted from DBpedia, we include plot keywords for movies from IMDb, represented as bag-of-words, i.e. each value is a string of keywords, not necessarily a single word, e.g. "human versus cyborg" is a keyword for "Terminator".

## 2.2 Ground truth labels

We propose tasks of varying cluster sizes in the GraB benchmark. That is, the difference in the datasets is in the cluster affiliations of nodes. Movie nodes are naturally grouped based on genre. However, propagating genre labels to person nodes requires some considerations. We devise the following three strategies to define label propagation and corresponding cluster notions.

- **Full affiliation:** The clusters of a movie are its genres. The actors, editors, producers, writers and directors inherit the genres of the movie. Intuitively, a person who worked on an adventure movie should be part of that cluster, even if said person has only worked on an adventure movie once.

- **Min affiliation:** Person nodes are only part of a cluster if they are affiliated with at least two movies of a given genre, unless a person is connected only to movies with unique genres, in which case we apply the *Full affiliation strategy*. As such, min affiliation removes some noisy labels from nodes with many different genres, but still affiliates all nodes with at least one cluster label.

- **Top-3 affiliations:** A person node is assigned the top three most frequent genre labels of its connected movies. In case of ties, we add to the node all the genres in the tie. This design choice favours popular genre affiliations.

The cluster structure varies with the design choice, as persons are affiliated with any genre they contribute to, repeated genre affiliations, or the most frequent genre affiliations.

## 2.3 Properties of GraB

The GraB graphs have 19 852 nodes with 67 843 features, 56 947 edges, and 22 genres.

**Cluster statistics (Table 2).** The standard deviation (std.) of the cluster sizes reveals that the size of the clusters varies considerably across all the datasets. The biggest cluster consists of Drama movies and affiliated persons, and the smallest cluster is Musicals (see also Fig. 2).

Two measures are used to describe the disjointedness of the graph, i.e. the number of nodes of a cluster unreachable from nodes of the same cluster. **Normal nodes** (NN) is the percentage of disjoint nodes. In case a node is part of two genres, e.g. both action and drama, and both action and drama are disjoint, we only count it as one node being disjoint. **Cluster nodes** (CN) is the percentage of nodes disjoint in all clusters. For instance, the node from the previous example counts as two disjoint nodes, one for action and one for drama.

**Min** and **Top-3** look similar except for NN and CN, which are lower in **Min** than in **Top-3**, indicating the majority of nodes are grouped with the rest of their cluster in the **Min** dataset, whereas in the **Top-3** dataset, more nodes of the same cluster are spread out.

**Degree & Density.** As all actors with only one edge have been removed, a few movie nodes have a single edge; less than 500 nodes have one edge. The number of nodes with two or more edges seems

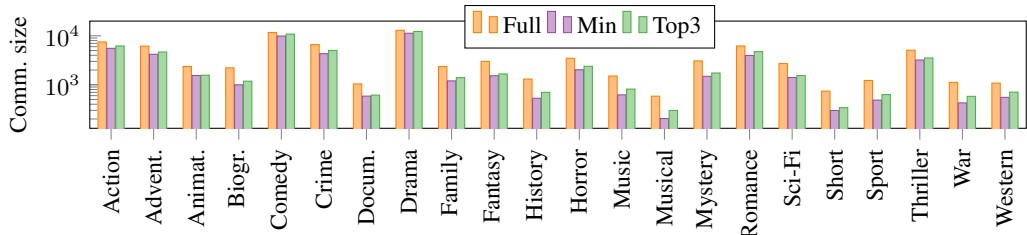

**Figure 2:** Commmunity sizes of GraB dataset.

to be exponentially decreasing, as shown in Figure 3 (log scale). The majority of the nodes have between two and five edges.

The density of the graph is 2.8e-4, meaning the graph is sparse. Graph clustering is harder on sparse graphs since the number of intra-cluster edges is not high, and the ratio between intra- and inter-cluster edges is low, posing a challenge for algorithms detecting clusters based on the graph structure [11]. Bear in mind, however, that this definition of intra- and inter-cluster edges does not apply to an overlapping cluster structure.

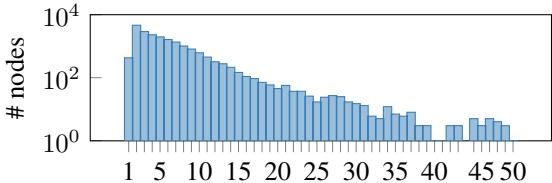

| | Any | All | Fractional | | | |
|---|---|---|---|---|---|---|
| | | | 25 | 50 | 75 | 100 |
| Full | 100 | 100 | 62.9 | 24.7 | 8.1 | 3.3 |
| Min | 96.6 | 75.6 | 74.2 | 35.3 | 11.9 | 8.3 |
| Top 3 | 92.8 | 40.3 | 77.4 | 52.6 | 12.9 | 12.9 |

**Figure 3:** Degree distribution of GraB (log scale)   **Figure 4:** Percentage of intra-cluster edges

**Inter- and intra-cluster edges.** We introduce three new measures *Any*, *All*, and *Fractional* of intra- and inter-cluster edges in an overlapping cluster structure setting. *Any* considers an edge as intra-cluster if the two connected nodes have at least one genre in common. *All* counts edges as intra-cluster if the labels of one node are a subset of the labels from the other node. In *Fractional*, we consider the Jaccard score between set of labels among two connected nodes in a cluster. If such a score exceeds a predefined threshold, the edge is intra-cluster. We set four thresholds, 0.25, 0.5, 0.75, and 1, resulting in edge statistics as in Figure 4. Edges that are not intra-cluster edges are, by definition, inter-cluster edges. A high percentage of overlapping intra-cluster edges should facilitate cluster discovery, as is the case for non-overlapping.

## 3 Empirical test of GraB

We empirically evaluate the challenges of GraB benchmark by running some common graph clustering algorithms. We test our datasets based on the quality of the evaluation of the algorithms, i.e. how similar is the predicted genres to the ground truth, with some common algorithms for graph clustering and algorithms using only graph structure or only attributes, respectively. This analysis provides a further argument for the hardness of our datasets.

- **Structure only:** *Spectral clustering (SC)* is an algorithm for non-overlapping graph clustering on non-attributed graphs. We use the `scikit-learn` implementation.

- **Attributes only:** *Expectation-maximisation (EM)* is an algorithm using attributes only to determine non-overlapping clusters. We use the `scikit-learn` implementation and a diagonal covariance to prevent memory overflows.

- **Graph Neural Networks (GNNs):** *DMoN* [12], *NOCD* [1], and *UCoDe* [13] are GNN algorithms for overlapping graph clustering.

We measure the clustering quality with ONMI [14] (Overlapping Normalized Mutual Information) and report the average and the max ONMI over 12 runs. For DMoN, NOCD, and UCoDe,

we report the average after training for 10 epochs, since we note no further improvement with more epochs. Table 5 shows the results of the experiments. We notice that the algorithms only using the structure or the attributes perform worse than the GNN algorithms, but the GNN algorithms still perform poorly. The performance of NOCD, DMoN, and UCoDe may improve if hyper-parameters are more finely tuned specifically for our

|  |  | SC | EM | DMoN | NOCD | UCODE |
|---|---|---|---|---|---|---|
| Full | A | 0.42 | 0.54 | 3.38 | 3.60 | 1.28 |
|  | M | 0.42 | 0.56 | 5.26 | 4.44 | 1.69 |
| Min | A | 0.63 | 0.23 | 2.68 | 0.84 | 0.77 |
|  | M | 0.63 | 0.23 | 4.1 | 1.05 | 0.98 |
| Top-3 | A | 0.054 | 0.23 | 0.8 | 0.38 | 0.68 |
|  | M | 0.054 | 0.23 | 1.5 | 0.54 | 0.91 |

**Figure 5:** Average (A) and max (M) ONMI results of common algorithms in percentage % on GraB benchmarks

datasets, but the overall performance level is not expected to change substantially. This could indicate that our datasets are challenging for different types of existing graph clustering approaches.

## 4    Conclusion

We propose GraB, a real-world benchmark for overlapping graph clustering in attributed heterogeneous graphs. We show that GraB is challenging for state-of-the-art graph clustering methods, including GNN-based ones. This indicates important open directions for future research, that can be benchmarked on GraB benchmark data. In future work, we plan to expand GraB further with more nodes and relationships, as well as include new attributes and define further ground-truth clusters.

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
