# OpenReview forum: "GraB: Graph Benchmark for Heterogeneous Graph Clustering"
_logconference.io/LOG/2022/Conference — LoG 2022 Poster_

### Official Review · Reviewer_uy6d · 2022-10-17

**Overall Score:** 3
**Confidence:** 4

**Review:**

## Main comment

In this article the authors propose a new benchmark for graph clustering. The new dataset is extracted from movies data that allow the authors to build a rich context with heterogeneity and class overlaps.

The approach is certainly interesting and such a benchmark could be useful to the community. I believe, however, that the article is not ready for publication yet. There are some choices and conclusions that appear to be rather subjective and that cannot constitute the basis a benchmark is built upon. The proposed graph must have a clear use and interpretation of its metadata.

A very big criticism I have (that I will further detail here below) is that the authors use a debatable way to define labels and they claim that their benchmark is interesting because none of the (very few) algorithms they tested is capable of retrieving those labels. It could well be that simply these labels are irrelevant or that the authors did not put enough effort in using state-of-the-art algorithms to solve the problem. After all, if we put labels at random, no algorithm could retrieve them, but that does not mean such a labeling could be of any use.

I will now get into deeper details with a point-to-point list of my concerns.

## Major concerns

* For a scientist a benchmark is a tool to test one's own work. As a consequence it should be as easy to use as possible. This dataset comes with no documentation beyond the way in which data should be uploaded. I think it would be necessary -- to improve both the clarity and the transparency of the article -- to include use examples in the GitHub page.

* On a related note, the authors test some algorithms on their benchmark dataset but do not provide the code to reproduce their results. I think this severly hampers the transparency of the authors claims and codes to reproduce the results should be provided both for pedagogical reasons and for clarity. This is particularly true when we think that two out of the three tested algorithms are for non-overlapping clusters. I tried to reproduce the results myself and got stuck simply because it is not clear what one should compare.

* I have a major philosophical issue with assigning genre labels to actors. While the procedure adopted to assign genre to movies is consistent and impartial, the one to assign a label to actors relies on an algorithm. We have data for movies, but not for actors and the authors provide three very simple strategies to associate a genre to each actor. Suppose, however that an algorithm to associate each actor to a genre is developed and that the output disagrees with the labels you provided. Why should we rely on your naive algorithm as the ground truth and admit that a more accurate algorithm fails in reconstructing the ground truth?

* Another, closely related issue, is that in this graph the authors represent people and movies at once, forming a bipartite graph. Moreover, among people, we have several roles mixed together but the labels only relate to genre. The structure of the graph is very likely to be much more determined by the type (movie or person) or the role of each node rather than the genre. The authors should, at least, provide all the labels that characterize each node.

* from line 140: the first two algorithms you test are meant for non-overlapping clustering. What is the purpose of testing an algorithm on a benchmark that has a different type on information to be retrieved? Moreover, the scikit-learn implementation of spectral clustering refers to articles that are approximately 20 years old. Several contributions have shown since then how to improve the results and performances of SC on real graphs. State-of-the-art methods should be adopted if one wants to claim that there is no algorithm that performs well.

## Minor concerns

* lines 9-12: graph clustering is one of the most widely explored problems in graph theory and it should be introduced with more diligence and references.

* line 16: well, real graphs are much more than just "sparse". They often are globally sparse (hence most nodes have a very small degree as compared to the graph size) but locally dense (there are a lot of triangles and small cliques, typically). On top of that, we might have community structures, broad degree distributions as well as an intimately multilayer structure so that the graph can be "sliced" seeing the interaction on different levels.

* be mindful when using the word "heterogeneous": heterogeneity can refer to many aspects of the graph and there is not a common use of this word. When using it, please clarify precisely what you mean.

* lines 23-24: " our focus is on providing benchmarking for such overlapping graph clustering as well". As far as I was able to test, it appears that this dataset provides a benchmark ONLY for overlapping graph clustering. All three strategies associate each actor to more than one label, hence, in all cases, clusters are overlapping. This lack of clarity is associated with the point already raised among the major concerns.

* Related works: the TUDataset and Open Graph Benchmark should be cited and commented.

* Table 1 seems to be inconsistent with the description given in the paragraph. It looks like all the datasets have overlapping clusters, which should not be the case.

* Please increase the white space after the caption on Figure 1.

* line 62: I do not think we can claim that a graph with approximately 20k nodes can be considered as large in these days. Graph with millions or even billions of nodes are publicly available.

* line 67: "We exclude person nodes with a single edge because they would only inherit the same
genres from that movie (as in Section 2.2), and trivially belong to the same clusters". Why would that be a problem?

* lines 73-81: this paragraph seems irrelevant to the ecology of the article. It would be much more appropriate to know precisely what features were used and how, rather than know that the formats are consistent that should be given for granted.

* line 86: "We propose tasks of varying cluster sizes in the GraB benchmark". This sentence is unclear.

* line 93: why do we make a distinction for people connected only to unique genres? This does not seem to be very consistent. I do not know if there is any instane in you dataset that aligns with my next example, but what would happen if an actor played only in movies with a single affiliation, being always the same? In that case adopting the "full affiliation" strategy would not make any difference.

* line 113: what does "look similar" mean? Be quantitative: I do not see any clear trend.

* line 120: replace the notation e-4 with 10^{-4} or what appropriate. Moreover, the degree/size ratio is not very telling to quantify sparsity. Suppose that the degree grows as n^{1/3} (where n is the graph size). Most researchers would agree that this is a rather dense graph, but for n sufficiently large n^{-2/3} can be very small in any case. The size of the graph and average degree (expressed individually) are better to quantify sparsity.

* Inter- and intra-cluster edges: what is the purpose of all this analysis? Please motivate it more deeply.

---

### Official Review · Reviewer_zdQe · 2022-10-18

**Overall Score:** 5
**Confidence:** 4

**Review:**

This paper provides a benchmark for overlapping clustering of a heterogeneous attributed graph created based on real data.

Pros:
P1- A review of the existing datasets for graph clustering in the literature (highlighted in Table 1) shows that existing datasets are either non-attributed, and/or non-heterogeneous, and/or created synthetically. Moreover, the size of some of the existing datasets is also small (<1000 nodes) limiting their use cases. From this perspective, the benchmark provided in this paper can fill a gap in the literature as it is based on real data, contains a  heterogeneous attributed graph, requires overlapping clustering, and the scale is larger than some of the existing datasets.
P2- The paper is very easy to read and follow.
P3- The fact that the algorithms based on node features and graph structure outperform the ones based on only one of these sources is encouraging and shows that both modalities are important.

Weaknesses:
W1- Looking at the results of different models in Table 3, it seems like all models are performing extremely poorly on all three versions of the dataset. This makes me wonder if this is because the dataset is challenging only for existing works (i.e. we can hope that future research can develop better methods that do well on this dataset), or because the dataset just does not contain the required information to be properly clustered (in which case, the dataset will be of limited use). Given that the main contribution of the paper is the dataset, further analysis is required to show if the low performance is due to the former or the latter.

W2- While the dataset is created based on real data, it is still a bit unclear in which real applications one may want to cluster a heterogeneous graph. This is especially unclear since nodes of different types (e.g., movies and actors) are being clustered together.

W3- The created dataset still seems quite small. Is there a reason why a bigger dataset is not created?

W4- I may be misunderstanding something but in the paragraph of lines 82-84, it is mentioned that bag of "keywords" is used (e.g., "human versus cyborg"). This makes me wonder how many movies these keywords appear in? e.g., if "human versus cyborg" appears only for one (or very few) movie, then the presence of that keyword is not going to be informative for clustering. This may relate to W1 as well (i.e. there may not be useful info in the graph for clustering to be done properly).

Suggestions for improvement:
S1- It would be great if the authors could find a way to report some notion of performance upper-bound (e.g., for an algorithm with infinite compute power and under some specific assumptions) to see how much improvement is possible using better clustering models. Alternatively, an analysis of why the existing models fail so miserably on the dataset can also be quite insightful (e.g., does the bi-partite nature of the graph make some algorithms fail?)
S2- It would be great if the authors add a few motivating real-world applications, where clustering a heterogeneous graph (especially when nodes of different types get clustered together) could be beneficial.
S3- Creating multiple versions of the dataset with different sizes could be useful.
S4- I suggest adding results for newer attributed graph clustering models in the next version (e.g., [1] -- table 3 -- reports better results than DMoN)

[1] Tackling Provably Hard Representative Selection via Graph Neural Networks

---

### Official Review · Reviewer_GpsX · 2022-10-21

**Overall Score:** 6
**Confidence:** 4

**Review:**

This paper introduces GraB, a new benchmark for heterogeneous graphs with overlapping clusters. I recommend acceptance for this benchmark because it targets a gap in the graph benchmarks landscape in which for this specific type of tasks, no real-world datasets existed before; furthermore, the empirical tests seem to show that better results can be developed by the community. In this sense, this work seems a very good fit for this conference, and the chosen movie domain to create this benchmark seemed a very good choice to start this specific type of real-world benchmarking.

I think the more obvious weakness with this benchmark is that this targets what seems to be a specific type of tasks that might not be used by a lot of people working with Graph ML, and only provides one dataset for benchmarking. However, this could be a good step forward to get more people interested, and therefore presenting this in this conference would clearly be the right place to do so.


Further weaknesses and comments:
1. In line 21 it is said that most of existing benchmarks focus on assigning nodes to a single cluster, but in table 1 it seems to be shown that the four benchmarks presented for comparison all have overlapping clusters. This seems to be a confusing mismatch in the paper.
2. In line 38 the concept "attributed graphs" is not defined, when there is an introduction section with lots of definitions in which this could be done.
3. In table 2, why the overlap size is only shown until 4, and there's not another column with, for instance, "4+"? I have the impression that bigger overlap sizes exist, and should therefore be documented.
4. I believe in line 139, when it is written "hardness", the authors meant "difficulty", or something similar.
5. I find the future work on expanding GraB very vague when it is said that they are planing on "increasing" the number of nodes, relationships, attributes and clusters. How exactly are the authors planing that? Do they have another dataset and domain in mind? Are they thinking on other ways to process DBpedia with other datasets beyond IMDb?
6. In lines 40-41 the paper mentions that bigger graphs allow for evaluation of scalability. However, this is not mentioned anywhere else in the paper, and would recommend that the authors revise this detail. In this sense, it might be useful for this benchmark to include hardware and inference time performance, for instance. It could be similar to things like MLPerf
7. Given the paper is an extended abstract in itself, I believe there is no need to have an abstract (as the whole paper is an abstract in itself). This could bring some extra space for the authors to, for instance, improve a bit the presentation of captions in figures/tables.

---

### Official Review · Reviewer_mGQe · 2022-10-22

**Overall Score:** 5
**Confidence:** 3

**Review:**

The paper presents GraB, a new data set containing graphs characterized by heterogeneous node types, node attributes, and can be used for clustering tasks with overlapping communities. The data set is built using information from IMDb and DBpedia.
After describing the construction of the graphs, with details on the nodes and their attributes and labels, the authors present the characteristic of the data set in terms of cluster statistics. Finally, the proposed data set is used for experiments and the clustering results are presented.

Strong points:
- publicly available data set for the data mining community
- GraB overcomes the limitations of existing datasets
- the description of the data set creation is clear

Weak points:
- some claims need to be clarified better (please see details comments)
- the empirical test could be improved
- the need for this type of data set needs to be motivated better
- the references need to be updated

Thanks for submitting to Log!
Based on the previous points, I think that the paper is not ready for acceptance in its current form.
In particular:
- From the introduction, it is not clear why it is important to have graphs with both node attributes and different types of nodes for overlapping clustering tasks. Adding another example (in addition to the one metabolic networks) would provide a stronger motivation for the paper.
-Line 22: "some recent approaches": which ones? please provide references.
-Line 41: "scalability": current clustering algorithms can run efficiently on graphs with hundreds of thousands of nodes and edges, the proposed data set seems not large enough to test scalability
-  The methods used in the experiments refer to ArXiv papers. Could you please provide the peer-reviewed and published versions of these methods? If they have not been published yet, I suggest considering using different approaches to have more reliable results and a  stronger experimental section.
- Could you please motivate how the results in the experiments support the claim that the data set is well designed? This is not clear to me. Probably more efficient methods even for clustering simple graphs or attributed graphs where the heterogeneous nodes and their connections could be interpreted as a multi-relational graph ([1] for instance and references therein) could provide completely different results.

[1] "Spectral Clustering of Attributed Multi-relational" Graphs in Proceedings of the 27th ACM SIGKDD Conference on Knowledge Discovery & Data Mining

---

### Meta-Review · Area_Chair_mfGs · 2022-11-09

**Confidence:** 4
**Recommendation:** Reject

**Meta Review:**

Reviewers all agree that the datasets provided by this work are crucial as a complementary testbed for the current graph ML community. However, there are a lot of valid concerns from reviewers, mostly centralized around 1) missing details about the datasets such as documentation, how to expand GraB, incomplete node attributes/labels, and so on, 2) missing insights into the results by running current methods on these proposed datasets. I strongly suggest the authors take the reviews seriously and do another-round revision.

---

### Decision · Program_Chairs · 2022-11-23

Accept (Poster)